# Effects of a Participation in a Structured Writing Retreat on Doctoral Mental Health: An Experimental and Comprehensive Study

**DOI:** 10.3390/ijerph20206953

**Published:** 2023-10-20

**Authors:** Cynthia Vincent, Émilie Tremblay-Wragg, Isabelle Plante

**Affiliations:** Department of Didactics, University of Quebec in Montreal, Montreal, QC H3C 3P8, Canada; tremblay-wragg.emilie@uqam.ca (É.T.-W.); plante.isabelle@uqam.ca (I.P.)

**Keywords:** doctoral researchers, mental health, wellbeing, writing retreats, mixed method study, experimental design

## Abstract

Challenges faced by doctoral researchers led to a concerning “doctoral mental health crisis” within academia. Recognizing the pressing need to address mental health concerns, notably among doctoral students, the Quebec Ministry of Higher Education introduced the Higher Education Student Mental Health Action Plan 2021–2026. One potentially relevant intervention approach is the implementation of tailored structured writing retreats for graduate students. Aiming to measure and explain the effects of participating to a three-day writing retreat on doctoral mental health, this study followed an explanatory sequential mixed method, including an experimental design. One hundred doctoral researchers were randomly assigned to either the experimental group (*n* = 50) or the waitlist control trial group (*n* = 50). Both groups answered a questionnaire comprising validated scales and open-ended questions at different timepoints, separated by a two-week gap. Results reveal that writing retreats reduced doctoral researchers’ psychological distress and improved their psychological, emotional, and social wellbeing. Among the multiple writing retreat aspects evaluated, only productivity experienced, as well as socialization/networking opportunities, acted as predictors for all doctoral mental health measures. Qualitative findings further supported the importance of perceived productivity and socialization/networking in promoting doctoral mental health. Recommendations are provided for fostering a supportive research work environment for doctoral researchers.

## 1. Introduction

In recent years, there was a growing recognition of the challenges faced by doctoral researchers, as their professional identity transforms from student to researcher [1]. Multiple systematic reviews emphasized the prevalence of mental health issues among this population, revealing a global ‘doctoral mental health crisis’ [2,3,4]. For instance, a survey of Belgian PhD students found that 51% of them experienced psychological distress, surpassing rates in other highly educated populations [5]. Similarly, studies in England showed low levels of doctoral wellbeing [6,7].

In Quebec, a Canadian province with numerous universities, the situation is similar. A large-scale survey by the Quebec Student Union (2019) [8] revealed that 51% of doctoral students reported high levels of psychological distress, findings that were associated with loneliness and lack of support. Therefore, the Quebec Ministry of Higher Education (2021) [9] published the Higher Education Student Mental Health Action Plan 2021–2026, which calls for evidence-based scientific evaluation of promising practices. To this end, the current study aims at measuring and understanding the effects of tailored writing retreats for graduate students as a potential intervention to enhance doctoral researchers’ occupational mental health.

## 2. Writing Retreats as a Mean to Improve Mental Health of Doctoral Researchers

Studies on doctoral mental health reveal that doctoral researchers face more tasks and responsibilities than ordinary employees. They often extend their working hours to evenings and weekends to assist or coordinate research projects, teach courses, earn a living, and work on their doctoral dissertation [5,10]. Their numerous work demands, such as assimilating scientific theories, generating ideas, managing data collection, data analysis, scientific writing, and reporting progress, can contribute to lower emotional wellbeing [11,12] and increased occupational stress [3,5]. 

According to literature review on interventions to improve doctoral researchers’ mental health by Mackie and Bates (2019) [13], positive psychology programs involving group meditation and mindfulness practices showed positive outcomes, with participants reporting increased wellbeing and reduced stress and anxiety [14]. Educating graduate students about physical activity and resilience was also effective in enhancing their understanding of positive mental health and self-care [15]. Finally, the implementation of a writing support group is yet to yield interesting results, with participants reporting reduced writing-related stress [16].

Similar to writing groups, structured writing retreats emerged as promising interventions for promoting doctoral researchers’ mental health. These retreats involve organized multi-day events where participants gather in a common space to write, share goals, and address challenges together [17]. Previous studies, using qualitative or quasi-experimental designs, explored the positive impacts of such retreats on writing productivity [18], writing self-regulation and self-efficacy [19], and writing flow [20]. Innovatively, Eardley et al. (2021) [21] investigated the potential of writing retreats as wellbeing interventions for academics. Employing a pre and post-test design without a control group, they found that structured writing retreats led academic participants to increased levels of emotional wellbeing. In-depth interviews revealed that this effect was mainly attributed to participants being able to prioritize writing over other job demands. Building on this work, the present study utilizes a rigorous experimental design, including both pre and post-tests, as well as a control group, to assess the effects of writing retreats on doctoral researchers’ occupational mental health—an understudied variable and population.

## 3. Mental Health in the Doctoral Research Context

The World Health Organization (2022) [22] defines mental health as a mental state functioning and existing on a complex continuum, rather than simply the absence of mental disorders. Accordingly, the dual-continua model of mental health posits the existence of negative mental health manifested with psychological distress, as well as positive mental health manifested with psychological, emotional, and social wellbeing [23,24]. In what follows, these various mental health components are addressed separately to offer a portrait of the specific literature pertaining to a doctoral context.

### 3.1. Doctoral Psychological Distress

Psychological distress, as an indicator of negative mental health, is a state in which a person experiences a variety of symptoms and internal experiences that are commonly considered troubling [25]. Accordingly, doctoral researchers’ psychological distress is described and measured as the presence of anxiety, depression, and stress, resulting from heavy research workloads [3,5]. Doctoral anxiety involves heightened apprehension, worry, and fear, often accompanied by feelings of restlessness and fear of academic failure [10]. Doctoral depression involves persistently feeling down, hopeless, and disinterested in doctoral activities that were once enjoyable. It can manifest itself in a lack of motivation and difficulty engaging or remaining focused on a doctoral task [26]. Lastly, doctoral stress is experienced in feelings of overwhelm, irritability, and physical symptoms such as agitation and tension due to academic expectations [3,10].

### 3.2. Doctoral Psychological Wellbeing

Psychological wellbeing, an indicator of positive mental health, encompasses self-acceptance, personal growth, purpose in life, environmental mastery, autonomy, and positive relations with others [24]. In the doctoral context, pursuing such a project signifies mastery in a specific research domain [4,27], driven by a strong sense of purpose and enjoyment of the intellectual and creative doctorate aspects [20,28]. Achieving the doctorate also requires a high degree of autonomy to overcome doctoral challenges [29] and self-acceptance to avoid imposter syndrome [27,30]. Consequently, psychologically well doctoral researchers feel healthy, energized, confident in their research tasks [26], and intellectually fulfilled [27].

### 3.3. Doctoral Emotional Wellbeing

Emotional wellbeing, or subjective wellbeing, is another component of positive mental health indicating a balance between positive and negative emotions, resulting in emotional stability and satisfaction [24,31]. Thus, doctoral emotional wellbeing is reflected in experiencing more positive emotions (e.g., happiness, joy, and satisfaction) than negative emotions (e.g., sadness, anger, and negativity) during research work [27,32,33].

### 3.4. Doctoral Social Wellbeing

Social wellbeing, another indicator of positive mental health and good functioning within society, encompasses social integration, contribution, coherence, actualization, and acceptance [24]. In the doctoral context, wherein students aim to become competent and recognized researchers, a crucial form of social wellbeing is the sense of scientific community [28]; that is, the perception of belonging to, influencing, and receiving support from the scientific community [34]. Belonging involves aligning with the academic culture, identifying with other scientists, and taking pride in belonging to the scientific community [35,36]. Influencing entails assuming a leadership role, making contributions, helping fellow members, and gaining recognition within the scientific community [37]. Lastly, receiving support involves various forms of assistance from other community members, including supervision [38], guidance [29], and academic support [34].

## 4. The Present Study

Overall, the available literature highlights the importance of doctoral mental health while underscoring the lack of evidence-based interventions to promote it. To address this gap, this experimental research project investigates the effects of writing retreats on doctoral researchers’ mental health. Furthermore, it aims to explore which aspects of this intervention are associated to such effects and investigate the underlying reasons for these associations. The study plan was preregistered at https://osf.io/38cbv (accessed on 22 December 2022).

## 5. Method

This study employs an explanatory sequential mixed method utilizing an experimental design where participants are randomly assigned to an experimental group (EG) and a waitlist control trial group (CG). The inclusion of a waitlist control group serves several important purposes [39]. Firstly, from an ethical perspective, it ensures that all participants, including those in the control group, have an opportunity to experience the benefits of the intervention. This approach aligns with principles of fairness and equity in research, reducing potential harm or disappointment for participants in the CG. Secondly, from a statistical standpoint, this methodology enhances the robustness of the study’s findings. By testing the replicability of the results through a second group undergoing the same intervention, it strengthens the confidence in the observed effects. If consistent results are obtained in both groups, it suggests that the findings are not merely due to chance or specific conditions, increasing the validity of the research outcomes. The mixed-method approach involves combining quantitative and qualitative data collection and analysis to interpret the writing retreats’ effects and gain a comprehensive understanding of the studied reality [40].

### 5.1. Intervention

The structured writing retreat intervention model used in this study is that of Thèsez-vous, a Canadian non-profit organization founded in 2015 to support graduate students in their academic writing [41]. With more than 100 editions, these retreats occur in various rural and suburban settings throughout the province of Quebec, predominantly in cost-effective and spacious spiritual centers conducive to quiet writing sessions. At a cost of CAD 300, the intervention includes lodging, structured activities, and meals, following the three-day schedule outlined in Figure 1.

According to Tremblay-Wragg et al. (2020) [42], the intervention model is inspired by Murray and Newton’s Structured Retreat Program for academics (2009) [43], incorporating three key principles: structuring the intervention, creating a “typing pool” environment, and fostering a community of practice. Accordingly, the intervention is structured with two facilitators guiding the group within the scheduled framework, introducing participants to the Pomodoro time management technique adapted for academic writing, which breaks down work into 50 min intervals of silent writing followed by a 10 min break. Facilitators also teach the specific, measurable, attainable, relevant, and time-bound (SMART) goal-setting method so participants can separate and narrow down their objectives. Ultimately, facilitators demonstrate the use of the giant Kanban board on the wall, a three-column board that serves to categorize participants’ SMART goals written on post-its into ‘to do’, ‘in progress’, and ‘completed’ work, allowing individual task monitoring and group progress visualization. Facilitators also encourage relaxation and engagement in recharging activities during breaks.

The “typing pool” concept involves scholars writing together in the same room for the entire retreat [43], fostering social engagement and creating an energizing environment that promotes writing flow [20]. The community of practice refers to the collective learning taking place around the activity of interest (writing) and the social relationships that are established during breaks [43]. The complete writing retreat process is outlined in Figure 2.

### 5.2. Participants and Procedures

The study sample comprised 100 doctoral researchers participating in one of 16 writing retreats offered between October 2022 and May 2023. After obtaining ethical approval from the authors’ institution (2023–4954), the primary researcher contacted colleges and universities in Quebec via email to inform them about the study and involve them in the recruitment process. Institutions were asked to promote the study by sending an email to their list of doctoral researchers and/or making a post on their social media page. Eligible participants included doctoral researchers from any university and research domain who either never attended a writing retreat or did so more than six months ago. Thus, the exclusion criteria were not being currently registered as a doctoral researcher, as well as having attended a writing retreat in the last six months. Those interested were invited to follow instructions and register for a writing retreat through Thèsez-vous’s website. Initially, 103 participants enrolled, but three were later excluded because they were master’s students. Participants were then randomly assigned to the waitlist control group (CG, *n* = 50) or the experimental group (EG, *n* = 50). Using LimeSurvey.com, online questionnaires were administered at specific timepoints preceding and following the intervention, as shown in Figure 3 (T1, T2, and T3), with a two-week gap between each timepoint.

This timeframe serves multiple purposes: it aims to counteract potential biases in participant responses, ensure consistency between groups, and capture the immediate effects of the intervention on mental health, in line with recommendations from Taber (2019) [44]. Response bias, common in short-term repeated questioning, occurs when participants become familiar with the questionnaire, remembering the questions and their previous answers, which may lead them to alter their responses. To address this, we implemented a two-week gap between the pre- and post-tests, minimizing the likelihood of participants recalling their earlier answers. Maintaining consistency in measurement timing between the control and experimental groups is crucial to accurately isolate intervention effects. Different measurement timings between the groups could introduce various uncontrolled events during these intervals, potentially leading to confounding variables that affect outcomes differently for each group [44]. By employing an identical two-week interval for both groups, we sought to minimize the influence of such confounding variables. Temporal bias considers how time influences study outcomes. Specifically, non-therapeutic and short interventions such as writing retreats are expected to yield immediate short-term benefits rather than gradual, long-term improvements. Furthermore, a delayed post-test could expose participants to uncontrolled external factors, impacting their mental health and responses [44]. Therefore, to capture immediate intervention effects while minimizing temporal bias, we conducted post-tests promptly after the writing retreat.

To ensure that a 72-hour assessment period for completing the questionnaires was respected, email reminders were sent to participants who did not respond within 24 h. As a result, this proactive follow-up procedure ensured that every participant completed their questionnaire within a maximum 60-hour timeframe. At T3, two participants from the waitlist control group did not attend the retreat and were subsequently ineligible to complete the last questionnaire. Participants in the waitlist control group received the intervention two weeks after their post-test. Therefore, all participants who experienced a writing retreat (*n* = 98) were included in the exploratory and qualitative data analyses, as such analyses do not require a control group. Table 1 provides information on participants’ characteristics across EG and CG at T1.

Table 1 shows that a higher proportion of participants were females, non-parents, in a relationship, and in the process of writing a dissertation in the social sciences. On average, participants were in their thirties and spent approximately 28 h per week working on their doctoral research, ranging from 4 to 60 h. In both groups, 31 out of 50 participants never previously participated in a writing retreat. Additional descriptive statistics showed that only 6 out of 100 participants had no other occupation besides their doctorate, while the majority were involved in assisting or coordinating research projects (67%), teaching courses (52%), performing other university tasks (47%), or working outside the university (42%). A total of 44 out of 100 participants held a provincial or federal dissertation grant. Furthermore, 19% of the sample had a mental health diagnosis, including anxiety (11%), depression (5%), post-traumatic stress disorder (2%), and attention deficit disorder (1%).

### 5.3. Measures

The questionnaire employed in this study included the four following validated French-language scales aiming to capture short-term changes in mental health state with sufficient sensitivity and specificity. At each timepoint, participants were instructed to evaluate the extent to which the statements applied to them over the past two weeks while they engaged in doctoral research activities (writing, analyzing data, etc.), ensuring the measures were contextualized. Additionally, a custom-made scale was developed to evaluate the perceived impact of writing retreat aspects on doctoral mental health post-intervention (i.e., at T2 for the EG and at T3 for the waitlist CG), and to gather qualitative interpretations from participants regarding these aspects.

#### 5.3.1. Doctoral Psychological Distress

Doctoral psychological distress was assessed using the Depression, Anxiety and Stress Scale—21 Items (DASS-21) developed by Lovibond and Lovibond (1995) [45] and available in 55 languages, including French, on their website (http://www2.psy.unsw.edu.au/dass/translations.htm, accessed on 17 July 2023). The DASS-21 includes three subscales measuring depression (α = 0.92), anxiety (α = 0.86), and stress (α = 0.90), each comprising seven items. Participants rated their experiences on a four-point Likert-type scale ranging from 0 (“did not apply to me at all”) to 3 (“applied to me most of the time”). Following the latest recommended guidelines, the subscale scores were summed to obtain a total psychological distress score, with a threshold of >16 indicating significant distress [46,47]. The DASS-21 provided high internal consistency in this study across all timepoints (α values ranging from 0.88 to 0.94).

#### 5.3.2. Doctoral Psychological Wellbeing

Doctoral psychological wellbeing was measured using the Doctoral Psychological Health Scale (DPHS) developed and validated in French by Vincent et al. (accepted) [26]. The DPHS is a unidimensional eight-item questionnaire that assess positive and negative psychological health (α = 0.91). Participants rated items on a five-point Likert-type scale from 1 (“very rarely or never”) to 5 (“very often or always”), and the mean score of the items provided a wellbeing indicator ranging from 1 to 5. The DPHS showed good internal consistency in this study across all timepoints (α values ranging from 0.87 to 0.91).

#### 5.3.3. Doctoral Emotional Wellbeing

Doctoral emotional wellbeing was measured using the Scale of Positive and Negative Emotion (SPANE) developed by Diener et al. (2009) [31] and available in French and 19 other languages on their website (http://labs.psychology.illinois.edu/~ediener/SPANE.html, accessed on 17 July 2023). The SPANE is a 12-item questionnaire with two subscales assessing positive emotions (SPANE-P, α = 0.87) and negative emotions (SPANE-N, α = 0.81), using a five-point Likert-type scale ranging from 1 (“very rarely or never”) to 5 (“very often or always”). The SPANE-P and SPANE-N scores were combined to calculate an overall affect balance score (SPANE-B, α = 0.89) ranging from −24 (unhappiest) to 24 (happiest). The SPANE-B demonstrated good internal consistency in this study across all timepoints (α values ranging from 0.88 to 0.93).

#### 5.3.4. Doctoral Social Wellbeing

Doctoral social wellbeing was assessed using the Sense of Scientific Community Scale (SSCS) developed and validated in French by Vincent et al. (2023) [26]. The SSCS is a three-subscale questionnaire with six items each, assessing the perception of belonging (ω = 0.92), influencing (ω = 0.91), and benefiting from support (ω = 0.97), using a five-point Likert-type scale ranging from 1 (“not at all”) to 5 (“completely”). The scores of the three subscales can also be summed to provide a total score (ω = 0.94). In the present study, SSCS Cronbach’s alpha values ranged from 0.95 to 0.96 across all timepoints.

#### 5.3.5. Writing Retreat Aspects

The impact of writing retreat aspects on doctoral mental health was measured post-intervention for all participants (*n* = 98) using a customized scale developed specifically for this study. Using a five-point bipolar scale (1—strong negative effect; 2—weak negative effect; 3—no effect; 4—weak positive effect; and 5—strong positive effect), participants were asked to evaluate the perceived effect of the six following writing retreat aspects on their mental health:Structured program animated by facilitators;quantity and quality of work produced;socialization and networking opportunities;writing and planning strategies employed;recharging activities undertaken;site, work facilities, and accommodations.

After each aspect assessed, an open-ended question asked participants to explain their response, thus generating qualitative data.

### 5.4. Analyses

Quantitative analyses were conducted using IBM^®^ SPSS Statistics, version 28. After initial data screening, Pearson’s correlations were used to examine the associations between the study variables. Main analyses included four separate two-way repeated-measures analysis of variance (ANOVA) to assess the potential effects of writing retreats on the four indicators of participants’ doctoral mental health. These ANOVAs included time (pretest/posttest) as a within-subject variable and group type (EG/CG) as a between-subject variable. Effect sizes were calculated using eta² (η²) and Cohen’s *d*, with guidelines from Van den Berg [48] (2022; η² = 0.01, small effect; η² = 0.06, medium effect; η² = 0.14, large effect); and Cohen (1988; *d*  =  0.2, small effect; *d*  =  0.5, medium effect; and *d*  =  0.8, large effect). Small, medium, and large effect size indicates that the intervention had a minimal, moderate, and substantial impact on the variability in doctoral mental health, respectively. To examine the replicability of the results, a second set of repeated-measure analyses was conducted, using the post-test scores of the waitlist CG as a pre-intervention measure, and the post-intervention scores as the post-intervention measure.

For the custom-made scale, a descriptive analysis was first conducted on each writing retreat aspect, evaluated by the 98 participants that experienced the writing retreat. Additionally, four regression analyses controlling for T1 values examined whether the writing retreat aspects acted as predictors of each of the four doctoral mental health indicators. Furthermore, a thematic qualitative analysis was conducted using Braun and Clarke’s (2006) [49] six-phase guidelines to gain a deeper understanding of participants’ perceptions regarding each retreat aspect and its effect on their doctoral mental health. The data corpus, comprising the responses to open-ended questions, was imported into Nvivo software version 1.7.1. The analysis involved familiarizing with the data, generating initial codes, searching for themes, reviewing potential themes, defining and naming themes, and producing the report (presenting each theme). A combination of inductive and deductive approaches was employed, with themes capturing patterns and meanings relevant to the research question (i.e., why did each writing retreat aspect impact participants’ mental health?).

## 6. Results

This section presents the results of each analysis described above separately.

### 6.1. Associations between the Study Variables

The correlation matrix (see Appendix A) revealed several significant associations among the variables. Overall, results show that mental health indicators are significantly intercorrelated, as expected. In addition, aside from age and having a mental health diagnostic, participant characteristics were not significantly correlated with the pre and post-test measures of doctoral mental health.

### 6.2. Effects of Writing Retreats on Doctoral Mental Health

Pre- and post-test mean scores are presented separately for CG and EG in Table 2.

As shown in Table 2, at pre-test, scores for CG and EG participants are quite similar, whereas, at post-test, scores for EG participants are lower for psychological distress and higher for psychological, emotional, and social wellbeing. Interestingly, at pre-test, mean scores surpass the threshold of 16, indicating the presence of psychological distress for both groups [46]. However, at post-test, only the CG still reaches the threshold. Results of the two-factor repeated measures ANOVA are reported in Table 3.

Results shown in Table 3 reveal significant Time × Group interactions for all variables (all *p* values < 0.006), with medium to large effects (all η_p_^2^ values < 0.08). Further decompositions of these interactions revealed no significant differences between the pre-test means of the CG and EG for all variables (all *p* values > 0.59, all *d* values < 0.11). In contrast, substantial variations emerged when comparing the post-test means of the CG and EG (all *p* values < 0.007, all *d* values > 0.68). Specifically, a remarkable decline in psychological distress and an important increase in doctoral psychological, emotional, and social wellbeing from the pre-test to the post-test were observed for the EG (all *p* values < 0.001, all *d* values > 0.71), in stark contrast to the CG (all *p* values >0.06, all *d* values < 0.38), as visually represented in Figure 4 below. Interestingly, while the positive effects of the retreat were moderate for doctoral social wellbeing (*d*  =  0.71), the effects on psychological distress (*d*  =  1.57) and emotional (*d*  =  1.84) and psychological (*d* = 1.55) wellbeing were notably larger. These findings provide strong empirical support regarding the benefits of writing retreats in improving various facets of doctoral researchers’ mental health.

To examine the replicability of the results, a second set of repeated-measure analyses was conducted, using the post-test scores of the waitlist CG as a pre-intervention measure and the post-intervention scores as the post-intervention measure. Results reveal non-significant interaction effects for doctoral psychological distress (*F*(1,96) = 2.00; *p* < 0.16; η_p_^2^ = 0.02), doctoral emotional wellbeing (*F*(1,96) = 0.97; *p* < 0.37; η_p_^2^ = 0.01), doctoral social wellbeing (*F*(1,96) = 0.38; *p* < 0.54; η_p_^2^ = 0.004), and doctoral psychological wellbeing (*F*(1,96) = 1.57; *p* < 0.21; η_p_^2^ = 0.02). In all four cases, the overall effect of time was significant, but not the overall effect of the group, suggesting that the waitlist CG derived similar benefits from the writing retreats as the EG. Overall, these additional analyses provide very similar results to those obtain with the main analysis previously presented, further confirming the positive role of writing retreats on doctoral mental health.

### 6.3. Perceived Effects of Writing Retreat Aspects on Doctoral Mental Health

Table 4 presents the descriptive results of participants’ evaluations regarding the effects of each writing retreat aspect on doctoral mental health.

On average, participants reported positive effects from the writing retreat on their doctoral mental health, with ratings ranging from 4.24 to 4.82. The data reveal a skewed distribution, with few participants indicating no effect or a negative effect on their mental health. Table 5 presents the results of the four regression analyses, examining the predictive role of writing retreat aspects on doctoral mental health indicators.

The regression results indicate that the predictors accounted for a significant amount of variance in doctoral mental health variables. For doctoral psychological distress, predictors explained 31.60% of variance (*F*(7,90) = 5.95, *p* < 0.001). Specifically, productivity (β = −0.29, *p* = 0.02) as well as socialization and networking (β = −0.23, *p* = 0.03) predicted less doctoral psychological distress. For doctoral psychological wellbeing, predictors explained 36.50% of variance (*F*(7,90) = 8.98, *p* < 0.001). Once again, productivity (β = 0.27, *p* = 0.02) as well as socialization and networking (β = 0.28, *p* = 0.01) were significant predictors of better doctoral psychological wellbeing. Similar results were observed for doctoral emotional wellbeing, with predictors explaining 41.70% of the variance (*F*(7,90) = 9.19, *p* < 0.001). Specifically, productivity (β = 0.42, *p* < 0.001) and socialization/networking (β = 0.25, *p* = 0.01) predicted higher emotional wellbeing. Lastly, for doctoral social wellbeing, predictors explained 52.50% of the variance (*F*(7,90) = 14.23, *p* < 0.001). This time, socialization and networking (β = 0.19, *p* = 0.04) was the only significant predictor of higher social wellbeing.

While participants generally rated all aspects of the retreat positively in terms of their contribution for mental health, the regression analyses highlighted the importance of productivity and socialization/networking as predictors of mental health outcomes. These findings suggest that these two factors may have a stronger influence on participants’ mental health compared to other aspects of the retreat. To gain further insights, the qualitative analysis of participants’ responses to the open-ended questions was conducted as part of the study’s sequential explanatory design.

### 6.4. Reasons Explaining the Perceived Positive Effects of Writing Retreat Aspects on Doctoral Mental Health

Table 6 summarizes the main qualitative findings on participants’ perceptions of each writing retreat aspect and its impact on their doctoral mental health. It includes an overview of each theme and examples of typical quotations illustrating positive and null/negative effects on mental health (translated from French to English).

As presented in Table 6, participants’ feedback provides valuable insights into their experiences and perspectives regarding each writing retreat aspect. Overall, results underscore that participants described the impact of productivity and socialization/networking in terms of mental health, while referring to other outcomes such as motivation or concentration for other aspects of the retreat.

## 7. Discussion

This explanatory sequential mixed method study sought to examine and understand the effects of participating in a writing retreat on doctoral researchers’ occupational mental health. Our study was among the first to use an experimental design, supplemented by a waitlist control trial group, to measure the effects of such an intervention in the field of higher education. The quantitative results demonstrate the beneficial effects of participating in a writing retreat in reducing doctoral psychological distress and enhancing doctoral psychological, emotional, and social wellbeing. The qualitative findings further shed light on the perceived explanations of these effects, specifically highlighting the role of productivity and socialization/networking experienced during the intervention. These findings carry substantial implications, paving the way for valuable recommendations to foster doctoral mental health.

### 7.1. The Positive Effects of Writing Retreats to Promote Doctoral Mental Health

Our results reveal that, while both groups were equivalent at the pre-test stage regarding all four examined variables related to doctoral mental health, the EG and CG significantly differed at post-test. This finding aligns with previous research highlighting the multiple potential benefits of focused writing environments [18]. In particular, our findings build upon the work of Eardley et al. (2021) [21], who observed an increase in emotional wellbeing among academics following a writing retreat. Our study confirms that writing retreats undeniably generate occupational emotional, social, and psychological wellbeing not only for academics, but also for doctoral researchers, while concurrently reducing psychological distress. Moreover, by breaking down doctoral mental health into multiple indicators, as recommended by recent studies on the assessment of mental health in doctoral researchers [27,50], the study was able to capture the writing retreat benefits from both negative and positive mental health perspectives, providing a more complete portrait of their effects.

### 7.2. The Positive Role of Writing Retreat Aspects to Promote Doctoral Mental Health

While participants positively rated all writing retreat aspects, only productivity and socialization/networking emerged as significant predictors of mental health outcomes. Qualitative analysis further supports these findings, with participants describing the impact of these two aspects on various mental health indicators. Indeed, on the one hand, productivity was reported to generate psychological relief, hope for the future, pride, and satisfaction, all indicators of doctoral psychological [27] and emotional wellbeing [28]. Conversely, participants experiencing difficulty in achieving desired productivity reported feelings of disappointment, self-depreciation, and anxiety, indicative of doctoral psychological distress [26]. Socialization and networking were reported to enhance positive emotions, reduce perceived distress by de-dramatizing doctoral challenges, as well as foster the sense of belonging to, benefiting from, and influencing the scientific community. Participants who negatively rated this writing retreat aspect reported feeling low fit with the academic profession, a known predictor of doctoral mental health problems [51]. These findings support previous assumptions: the possibility to connect around common interests and similar challenges during writing retreats [18] apparently fosters a positive effect [21] and relational wellbeing [52]. 

On the other hand, the writing pool was primarily seen as promoting concentration, as demonstrated previously [19], rather than directly impacting mental health indicators. Adopting writing and planning strategies appeared to increase perceived productivity and self-efficacy through goal-setting and planning, consistent with Vincent et al.’s (2021) [19] findings. However, some participants did not find certain techniques, such as SMART objectives and Kanban boards, useful, perceiving them as time wasting. Recharging activities, particularly nature walks, were described as resourcing and stimulating deeper thought and engagement, in line with prior research [53]. While most participants recognized the importance of these breaks in replenishing energy, some viewed them as interruptions to productivity due to the competitive research culture, where time spent on self-care is undervalued [54]. Lastly, the site and all-inclusive package were perceived as facilitators of productivity by relieving participants of additional responsibilities, such as cooking and cleaning, therefore maximizing working time [18,41]. Overall, participants did not describe these aspects as likely to affect their mental health, but rather their organization and focus, which may explain why they did not act as predictors in the multiple regression analysis.

Nonetheless, qualitative data uncovered nuanced aspects of participants’ experiences that quantitative data could not capture. Notably, not everyone may benefit from the same aspects of the retreat. It is worth considering the possibility that distinct participant profiles exist, each with their own motivations for attending. This variety in participants’ experiences and motivations suggests that a one-size-fits-all approach to retreat interventions may not be optimal. Additionally, qualitative findings suggest interrelatedness among retreat aspects, where increased motivation and concentration contributed to enhanced productivity and facilitated socialization/networking. In order to examine the generalizability of these qualitative observations, future research using structural equation modeling could explore the complex relationships between doctoral motivation, concentration, productivity, socialization/networking, and mental health outcomes.

### 7.3. Practical Implications

Study findings hold significant practical implications for doctoral researchers and educational institutions, offering insights for managing work–life balance and designing better work and social spaces for student researchers [54]. To begin, because of the strength of the benefits obtained in this study, successful initiatives such as Thèsez-vous or similar writing retreat programs should be promoted and implemented. To enhance positive mental health outcomes, these interventions, whether organized by students, organizations, or academic institutions, should encourage productivity and opportunities for socialization and networking with key principles, such as structuring the intervention with facilitators and techniques such as Pomodoro, SMART goals, and Kanban. Additionally, fostering a collaborative “typing pool” environment and organizing socialization/networking activities can cultivate a sense of community and promote shared learning. Academic institutions should also consider the potential benefits of incorporating initiatives such as writing events and dedicated writing spaces into doctoral training and professional development.

Furthermore, it seems crucial to acknowledge that doctoral researchers often feel that mental health services and programs at their university are not tailored to their needs, and consequently, are a waste of time, primarily because they are not embedded into the research culture that requires juggling different time-consuming responsibilities [55,56]. Therefore, existing support programs and university initiatives should recognize the central role of productivity as a potential catalyst of doctoral wellbeing. Consequently, they could align their efforts accordingly to provide resources and guidance on effective time management techniques and goal-setting strategies. Likewise, because of the positive role of socialization/networking to promote mental health, fostering a favorable environment that supports doctoral researchers’ social connectedness across disciplines could enhance the effectiveness and relevance of their support initiatives [57]. Importantly, using such preventive approaches aligns with recent recommendations, as they are likely to prevent mental health problems and program dropout [55].

## 8. Limitations and Future Research

Several limitations should be acknowledged. First, the participants, being doctoral researchers recruited mainly by higher education institutions promoting the study through emails and social media who voluntarily registered for a writing retreat, may possess different characteristics or motivations compared to other doctoral students, potentially introducing a selection bias. However, the recruitment of a sample including 62% of participants who never attended a writing retreat before mitigates this concern and contributes to the study’s representativeness. Second, the gender composition of the sample was skewed, with only 24% identifying as men. This ratio does not align with the national gender distribution at the doctoral level across disciplines in Canada, where men typically outnumber women, except in education and psychology [58], two disciplines that many of our participants were enrolled in. Consequently, the study faced limitations when attempting to conduct quantitative analyses of potential gender differences. For instance, the study could not corroborate the well-known observation that women doctoral researchers experience worse mental health states than men during their studies [3,5], which is likely due to insufficient statistical power. As a result, the study may not fully capture potential gender-specific experiences and challenges related to mental health in the doctoral research context. Despite the common occurrence of gender bias in health behavior research [59], addressing this limitation with more representative samples in future studies is essential for enhancing the generalizability of the results and for better understanding gender inequalities in mental health among doctoral researchers.

Third, the study relied on self-report measures for assessing mental health outcomes, which may be influenced by factors such as social desirability bias or subjective interpretations of the items. Incorporating objective measures or additional data sources, such as observations by clinical professionals, could provide a more comprehensive understanding of participants’ mental health. Finally, the study focused on short-term effects and did not assess long-term outcomes, leaving the sustainability of observed improvements in mental health unclear. Future research should include follow-up assessments to determine the durability of effects and explore factors influencing the maintenance of improved mental health. Investigating the role of individual characteristics, facilitator qualities, and different types of retreats (e.g., virtual retreats, discipline-specific retreats) could be potential avenues for future research.

## 9. Conclusions

In conclusion, this study provides evidence of the positive effects of participating in a writing retreat on doctoral researchers’ mental health. The findings highlight the importance of productivity gains and socialization/networking experiences in driving these positive outcomes. These results can inform the design of more effective interventions to support the mental health and wellbeing of doctoral researchers, aligning with the goals of academic institutions and policymakers seeking to enhance mental health support services in universities [9].

## Figures and Tables

**Figure 1 ijerph-20-06953-f001:**
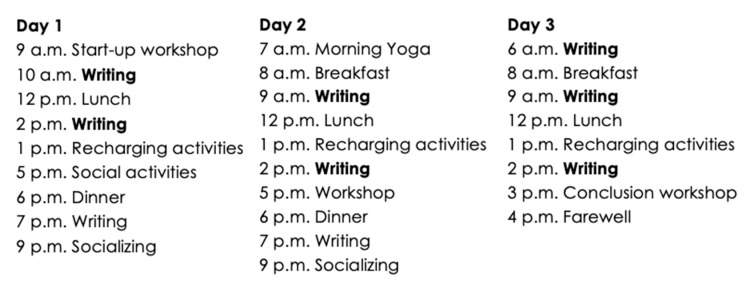
Thèsez-vous writing retreats schedule (Tremblay-Wragg et al., 2020).

**Figure 2 ijerph-20-06953-f002:**
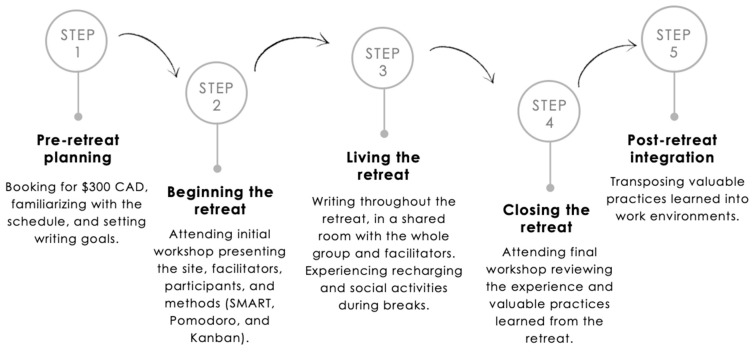
Thèsez-vous structured writing retreats steps for participants (inspired from Vincent et al., 2021).

**Figure 3 ijerph-20-06953-f003:**
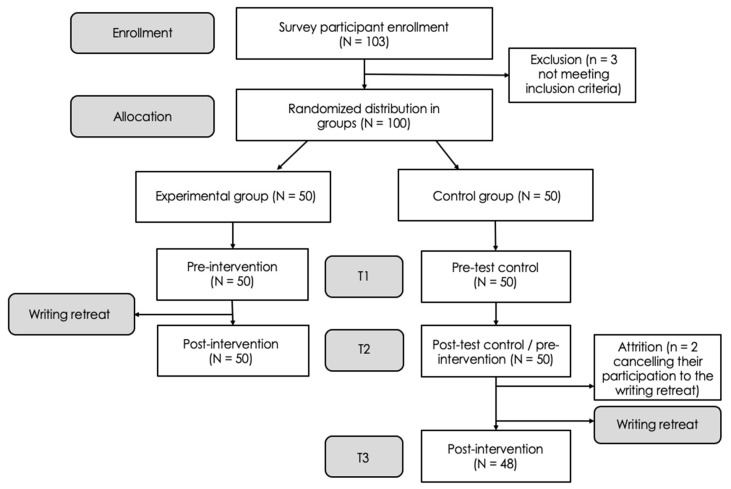
Flowchart describing the study procedure.

**Figure 4 ijerph-20-06953-f004:**
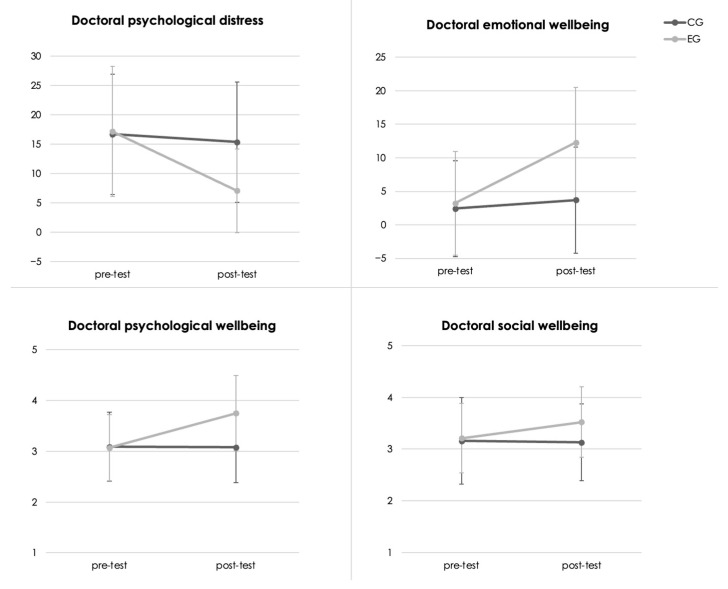
Illustration of writing retreat effects on doctoral mental health indicators.

**Table 1 ijerph-20-06953-t001:** Participants’ characteristics at T1.

Demographics	Experimental Group	Control Group
Participants	50	50
Age mean (SD)	33.48 (6.81)	32.40 (4.64)
**Participant gender**		
Female	36	39
Male	13	11
Non-binary	1	0
**Familial status**		
Nonparents	39	36
Parents	11	14
Single	12	10
In a relationship	38	40
**Doctoral stage**		
Tuition	9	11
Research project	11	7
Dissertation writing	30	32
**Research domain**		
Social Sciences	40	40
Health, pure, and applied sciences	10	10
Hours worked/week mean (SD)	28.94 (13.48)	27.16 (12.71)
Never attended a writing retreat	31	31

**Table 2 ijerph-20-06953-t002:** Descriptive statistics.

	Pre-Test			Post-Test			Waitlist Post-Intervention		
Group	*n*	Score	SD	*n*	Score	SD	*n*	Score	SD
Doctoral psychological distress									
CG	50	16.58/63	10.28	50	15.56/63	10.25	48	8.69/63	8.70
EG	50	17.16/63	11.07	50	7.08/63	7.14			
Doctoral psychological wellbeing									
CG	50	3.09/5	0.68	50	3.08/5	0.70	48	3.74/5	0.65
EG	50	3.07/5	0.65	50	3.75/5	0.74			
Doctoral emotional wellbeing									
CG	50	2.42/24	7.16	50	3.72/24	7.91	48	10.81/24	7.27
EG	50	3.22/24	7.73	50	12.28/24	8.24			
Doctoral social wellbeing									
CG	50	3.16/5	0.84	50	3.13/5	0.74	48	3.34/5	0.75
EG	50	3.21/5	0.67	50	3.52/5	0.68			

**Table 3 ijerph-20-06953-t003:** Global effects.

Source	*F*	*p*	η_p_^2^
Doctoral psychological distress			
Time	38.72	<0.001	0.28
Group	4.85	0.03	0.05
Time × Group	23.80	<0.001	0.20
Doctoral psychological wellbeing			
Time	46.45	<0.001	0.32
Group	3.57	0.06	0.04
Time × Group	17.31	<0.001	0.15
Doctoral emotional wellbeing			
Time	59.75	<0.001	0.38
Group	11.15	0.001	0.10
Time × Group	33.52	<0.001	0.26
Doctoral social wellbeing			
Time	5.14	0.026	0.05
Group	2.70	0.10	0.03
Time × Group	7.90	0.006	0.08

Note. *df*(1.98) for all variables.

**Table 4 ijerph-20-06953-t004:** Descriptive statistics of the perceived effects of the writing retreat aspects on doctoral mental health (*n* = 98).

Items	Mean	SD	Skewness	Kurtosis
Writing pool (gathering in the same room to work, managed by facilitators)	4.82/5	0.58	−4.16	20.89
Productivity (quantity and quality of work done)	4.66/5	0.63	−2.20	5.62
Site (work facilities and accommodations)	4.61/5	0.65	−1.91	4.09
Writing and planning strategies	4.56/5	0.68	−1.26	0.30
Socialization and networking	4.35/5	0.76	−1.11	1.04
Recharging activities	4.24/5	0.91	−1.01	0.50

**Table 5 ijerph-20-06953-t005:** Regression analysis results (*n* = 98).

	Psychological Distress ^a^		Psychological Wellbeing		Emotional Wellbeing		Social Wellbeing	
Predictors	β	*p*	β	*p*	β	*p*	β	*p*
Control T1 ^b^	0.33	0.00	0.44	0.00	0.40	0.00	0.59	0.00
Writing pool	−0.07	0.59	0.10	0.38	−0.03	0.80	0.14	0.16
Productivity	−0.29	0.02	0.27	0.02	0.42	0.00	0.09	0.36
Socialization and networking	−0.23	0.03	0.28	0.01	0.25	0.01	0.19	0.04
Writing and planning strategies	−0.12	0.23	0.08	0.37	0.06	0.48	0.02	0.82
Recharging activities	−0.08	0.44	0.04	0.70	0.01	0.89	0.12	0.16
Site	0.20	0.06	−0.17	0.08	−0.17	0.09	−0.17	0.06

Note: ^a^ To account for outliers, logged scores for psychological distress were used in the regression analysis predicting this outcome. ^b^ For each regression analysis predicting one of the four mental health indicators, scores for this indicator at T1 were entered as a control variable.

**Table 6 ijerph-20-06953-t006:** Results of qualitative analysis *(n* = 98).

Theme	Quotation for Positive Effect on Doctoral Mental Health	Quotation for Null or Negative Effect on Doctoral Mental Health
Writing pool	*n =* 94	*n =* 4
Created a group effect conducive to writing concentration	Gathering in the same room allowed me to feel driven by the group’s energy. Time management by the facilitators is essential for this group energy, because during the 50-minute work session, everyone works, and no one disturbs or procrastinates on social networks. So, [facilitators] provide a discipline that encourages concentration. (CG50)	The group effect didn’t particularly stimulate me. Time management was rather disturbing for my concentration. (EG6)
Productivity	*n =* 94	*n =* 4
Provided psychological relief, pride, satisfaction, and hope for the future.	I’ve accomplished more in three days than I have in the last two weeks! Making such progress in my work removes a great deal of anxiety about my ability to achieve my objectives and deadlines. I feel accomplished and proud of what I’ve been able to achieve. (EG14)	I came up with goals that weren’t achievable. […] So, I leave the retreat with “what I should have done” instead of thinking “wow, I’ve done so much”. (EG43)
Socialization and networking	*n =* 87	*n =* 11
Enhanced positive emotions as well as sense of belonging, benefiting, and influencing the scientific community.	I realized I wasn’t the only one who was stressed and negative. We laughed about our concerns and challenges, rather than seeing them negatively because we think we’re alone in experiencing them. The people around me don’t understand the challenges I face. The other doctoral researchers understand! I met people working on subjects related to mine too, so it gave me a renewed passion for my subject. (CG34)	As my career progresses, I feel less and less a part of the scientific community. The clinical environment appeals to me more and I feel that I belong more there. [At the retreat], finding myself only with researcher students talking about their dissertation and academic background exacerbated my feeling of not fitting in. (EG9)
Writing and planning strategies	*n =* 88	*n =* 10
Enhanced perceived productivity and self-efficacy.	The tools and working techniques enhance efficiency considerably. The 50-min Pomodoro is ideal to focus when we’re writing. And to see everyone’s objectives moving throughout the retreat [through the giant Kanban board] is highly motivating. (EG47)	I don’t feel the need to break my objectives down into small tasks. When I set to work, I know what I have to do and I do it. It would be a waste of time for me to write down what I have to do. (EG24)
Recharging activities	*n =* 76	*n =* 22
Provided an opportunity to unwind and refresh.	I loved being able to enjoy the river and the surrounding forest trails through walks or runs. Each time, I felt re-energized to engaged into another pomodoro. (EG2)	I didn’t do such activities. I was there to work. (EG34)
Site and all-inclusive package	*n =* 93	*n =* 5
Helped boost productivity.	Working in a large, quiet, and well-lit room with a magnificent view clearly contributed to my productivity and writing flow, as did being fed without the mental burden of cooking, planning meals, and cleaning up dishes. Everything was optimal. (CG35)	The working facilities could have been a little more ergonomic, the chairs weren’t the most comfortable and the tables weren’t at the right height, but for just three days, it works. (CG33)

## Data Availability

The data presented in this study are available on request from the corresponding author.

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
