# Peer review of "Effects of a Participation in a Structured Writing Retreat on Doctoral Mental Health: An Experimental and Comprehensive Study"

_ijerph, 2023, doi:10.3390/ijerph20206953_

Round 1

Reviewer 1 Report

This article addresses the theme of the challenges faced by doctoral researchers, which have contributed to a concerning "doctoral mental health crisis" within academia. This is a relevant and interesting topic for the journal's readers.

A noteworthy aspect of the research is the authors' use of an experimental design. This is a robust approach to assess the effects of an intervention, such as writing retreats, on the mental health of doctoral researchers. It helps establish causal relationships.

However, there are some aspects that could be further elaborated by the researchers:

Clarify that there may be a recruitment bias among the participants. The recruitment of participants appears to rely primarily on higher education institutions promoting the study through emails and social media. This could lead to selection bias, as participants who chose to enroll in the writing retreats may not be representative of all doctoral candidates.

The text mentions that three participants were excluded because they were master's students, which is a reasonable decision. However, it's unclear if other exclusion criteria were applied, and it would be important to know if there were additional criteria beyond this.

The two-week interval between different data collection points may be appropriate, but it's essential to consider whether this interval is sufficient to capture the effects of the writing retreat on doctoral researchers' mental health. Additionally, it's not clear if participants were followed up after the retreat for a longer period to assess long-term effects.

While the described procedures have some limitations and considerations that need to be addressed, the research holds significant merit, and the authors are to be commended for their work.

Author Response

  1. Clarify that there may be a recruitment bias among the participants. The recruitment of participants appears to rely primarily on higher education institutions promoting the study through emails and social media. This could lead to selection bias, as participants who chose to enroll in the writing retreats may not be representative of all doctoral candidates.

Thank you for pointing this out. We agree with this comment. Therefore, we have mentioned this selection bias in the Limitations and Future Research section: “First, the participants, being doctoral researchers recruited mainly by higher education institutions promoting the study through emails and social media and who voluntarily registered for a writing retreat, may possess different characteristics or motivations compared to other doctoral students, potentially introducing a selection bias.” (p.16, l.545-549).

  1. The text mentions that three participants were excluded because they were master's students, which is a reasonable decision. However, it's unclear if other exclusion criteria were applied, and it would be important to know if there were additional criteria beyond this.

We concur with this comment. “Eligible participants included doctoral researchers from any university and research domain, who had either never attended a writing retreat or had done so more than six months ago. Thus, the exclusion criteria were not being currently registered as a doctoral researcher, as well as having attended a writing retreat in the last six months.” (p.5, l.198-202)

  1. The two-week interval between different data collection points may be appropriate, but it's essential to consider whether this interval is sufficient to capture the effects of the writing retreat on doctoral researchers' mental health. Additionally, it's not clear if participants were followed up after the retreat for a longer period to assess long-term effects.

Thank you for pointing this out. To clarify why we chose this interval and how it is made to capture the effects of the writing retreat on doctoral researchers' mental health, we added this section (p.5-6, l.210-228):

“This timeframe serves multiple purposes: it aims to counteract potential biases in participant responses, ensure consistency between groups, and capture the immediate effects of the intervention on mental health, in line with recommendations from Taber (2019). Response bias, common in short-term repeated questioning, occurs when participants become familiar with the questionnaire, remembering the questions and their previous answers, which may lead them to alter their responses. To address this, we implemented a two-week gap between the pre- and post-tests, minimizing the likelihood of participants recalling their earlier answers. Maintaining consistency in measurement timing between the control and experimental groups is crucial to accurately isolate intervention effects. Different measurement timings between the groups could introduce various uncontrolled events during these intervals, potentially leading to confounding variables that affect outcomes differently for each group (Taber, 2019). By employing an identical two-week interval for both groups, we sought to minimize the influence of such confounding variables. Temporal bias considers how time influences study outcomes. Specifically, non-therapeutic and short interventions like writing retreats are expected to yield immediate short-term benefits rather than gradual, long-term improvements. Furthermore, a delayed post-test could expose participants to uncontrolled external factors, impacting their mental health and responses (Taber, 2019). Therefore, to capture immediate intervention effects while minimizing temporal bias, we conducted post-tests promptly after the writing retreat.”

We hope that it also clarifies that participants were followed up immediately after the retreat, since it is a non-therapeutic and short interventions to assess short-term effects and not for a longer period to assess long-term effects. Note that we also acknowledge this possibility that in the Limitations and Future Research section:

the study focused on short-term effects and did not assess long-term outcomes, leaving the sustainability of observed improvements in mental health unclear. Future research should include follow-up assessments to determine the durability of effects and explore factors influencing the maintenance of improved mental health.” (p.16, l.570-572)

Since the expected time for participants to experience changes in mental health may vary, not only according to the nature of the intervention, but also according to the mental health criteria measured, we have specified in the measurement section:

“The questionnaire employed in this study included four validated French-language scales aiming to capture short-term changes in mental health state with sufficient sensitivity and specificity.” (p.7, l.257-259)

  • 4. While the described procedures have some limitations and considerations that need to be addressed, the research holds significant merit, and the authors are to be commended for their work.

We thank you for your encouraging comments and highly beneficial insights.

Reviewer 2 Report

This manuscript describes the results of a randomized controlled trial (with waitlist controls, 50 v 50) to investigate the effects of participation in a structured three-day writing retreat on the mental health and wellbeing of doctoral students (four measures). As well as the experimental phase of the trial, additional quantitative analyses are reported for all participants completing the retreat to assess the impact of six perceived aspects of the retreat on outcomes. Further, qualitative analyses are summarised for individually perceived positive and negative components of the retreat. The main findings from the experimental design were that participants rated themselves significantly better on psychological distress, psychological wellbeing, emotional wellbeing, and social wellbeing compared to waitlist controls. Regression analyses emphasised the roles of (1) productivity and (2) socialization and networking from the six evaluated components as impacting on outcomes. The qualitative analyses were informative in illustrating individual differences around the general outcomes based on quantitative analyses. These may also inform refinements of future versions of the retreat.

The strengths of the study include the experimental design, the selected outcome measures, the enhancement to investigate most salient aspects of the retreat, and the mixed-method approach. Of course, these necessarily rested on Thèsez-vous and the SMART goals of the intervention model. The authors point out the limitation of assessing only short-term outcomes (an accepted constraint of using waitlist controls) and they discuss the limitation that only volunteers participate in the retreats. Overall, the study is very well conducted and presented. None of the following comments are of a perilous nature and are offered to assist with revisions.

1)      I note from the pre-registered study design that one of the hypotheses was “It is expected that an examination of gender differences will reveal a greater improvement in psychological health and sense of scientific community among women than among men participating in a writing retreat.” It is possible that the researchers have further plans to explore this issue and also possible that the recruitment of less that 30% male students has stretched the power of the study too far. When covering this in the limitations, could a little more be said about how biased this recruitment is (e.g. is the target population actually 50:50)? More generally, are there other known biases in those who volunteer for the retreats?

2)      The formatting of the first column of Table 1 is a bit off (looks like the entries are centred rather than left justified). Numbers for sub-groups would be easier to follow if this could be rectified.

3)      I couldn’t quite follow on page 7 that the custom-made scale was for “the perceived impact of writing retreat aspects on doctoral mental health at T3”. It seems to have been used for participants in both arms of the trial so it should be made more clear that it was used at T2 or T3 depending on group membership.

4)      There is a comment in the Discussion about “recruitment of individuals who had never attended a writing retreat before” but I could find no mention of this in the Results, only that the Method specified participants “either never attended a writing retreat or had done so more than six months ago”. If this is relevant, the information should be presented.

5)      The instructions on how participants completed the key outcomes are not mentioned. Usually such measures have a time period for self-assessment, e.g. 7 days, but this seems less relevant to the short-term effects involved in a three-day retreat. What period of assessment was used for the outcomes?

6)      The Results section contains plenty of detail for those familiar with the statistics routinely employed in psychology but less for those with backgrounds in epidemiology and public health. Can the authors explain more about the effect sizes attributable to the intervention and their significance in practical terms?

7)      I can see why the authors wait until page 10 to elaborate on the second repeated measures analysis (using the intervention measures of the “waitlist control group”). I’m not so sure all readers will grasp this on first read and wonder if it merits a few more words to explain what was done and why it was important. (The authors already explain that these results reinforce the outcomes of the initial analysis.)

8)      The qualitative findings seemed a little underplayed in the Discussion. One thing such results illustrate is that not everyone falls neatly into the general pattern indicated by quantitative findings. Is this worth a comment? Public health research is awash with indices of central tendency but real people have a funny way of not keeping in step.

9)      Btw, I couldn’t get through to the Diener site via the link in the manuscript, and two of the three questionnaire links on the OSF registry returned blanks. Maybe it’s just a bad internet day but it might be worth checking those links again.

It was a pleasure to read a careful and meaningful study.

Author Response

1. I note from the pre-registered study design that one of the hypotheses was “It is expected that an examination of gender differences will reveal a greater improvement in psychological health and sense of scientific community among women than among men participating in a writing retreat.” It is possible that the researchers have further plans to explore this issue and also possible that the recruitment of less that 30% male students has stretched the power of the study too far. When covering this in the limitations, could a little more be said about how biased this recruitment is (e.g. is the target population actually 50:50)? More generally, are there other known biases in those who volunteer for the retreats?

We have added such information in the Limitations and Future Research section:

“Second, the gender composition of the sample was skewed, with only 24% identifying as men. This ratio does not align with the national gender distribution at the doctoral level across disciplines in Canada, where men typically outnumber women, except in education and psychology (Deng, 2021), two disciplines that many of our participants were enrolled in. Consequently, the study faced limitations when attempting to conduct quantitative analyses of potential gender differences. For instance, the study could not corroborate the well-known observation that women doctoral researchers experience worse mental health states than men during their studies (Hazell et al., 2020; Levecque et al., 2017), likely due to insufficient statistical power. As a result, the study may not fully capture potential gender-specific experiences and challenges related to mental health in the doctoral research context. Despite the common occurrence of gender bias in health behavior research (Ryan et al., 2019), addressing this limitation with more representative samples in future studies is essential for enhancing the generalizability of the results and for better understanding gender inequalities in mental health among doctoral researchers.”(p.16, l.551-564)

2. The formatting of the first column of Table 1 is a bit off (looks like the entries are centred rather than left justified). Numbers for sub-groups would be easier to follow if this could be rectified.

Thank you for pointing this out. The first column of Table 1 is now left justified.

3. I couldn’t quite follow on page 7 that the custom-made scale was for “the perceived impact of writing retreat aspects on doctoral mental health at T3”. It seems to have been used for participants in both arms of the trial so it should be made more clear that it was used at T2 or T3 depending on group membership.

We clarified that point by adding the following precisions in the Measures section: “Additionally, a custom-made scale was developed to evaluate the perceived impact of writing retreat aspects on doctoral mental health post-intervention (i.e. at T2 for the EG and at T3 for the waitlist CG), and to gather qualitative interpretations from participants regarding these aspects.” (p.7, l.262-265)

“The impact of writing retreat aspects on doctoral mental health was measured post-intervention for all participants (N = 98) using a customized scale developed specifically for this study.” (p.8, l.306-307)

4. There is a comment in the Discussion about “recruitment of individuals who had never attended a writing retreat before” but I could find no mention of this in the Results, only that the Method specified participants “either never attended a writing retreat or had done so more than six months ago”. If this is relevant, the information should be presented.

As suggested, the information is now properly presented in our manuscript. We have appropriately incorporated the results regarding the number of individuals who had never attended a writing retreat before in both groups into Table 1. We also provided a clear textual description bellow Table 1: “In both groups, 31 out of 50 participants had never previously participated in a writing retreat.” (p.7, l.247-248).

To provide even more transparency, we also restated the result in the Limitations and Future Research section: “However, the recruitment of a sample including 62% of participants who had never attended a writing retreat before mitigates this concern and contributes to the study's representativeness.” (p.16, l.549-550).

5. The instructions on how participants completed the key outcomes are not mentioned. Usually such measures have a time period for self-assessment, e.g. 7 days, but this seems less relevant to the short-term effects involved in a three-day retreat. What period of assessment was used for the outcomes?

The period of assessment to complete the questionnaires was clarified in the Participants and Procedures section: “To ensure that a 72-hour assessment period for completing the questionnaires was respected, email reminders were sent to participants who had not responded within 24 hours. As a result, this proactive follow-up procedure ensured that every participant completed their questionnaire within a maximum 60-hour timeframe.” (p.6, l.229-232).

6. The Results section contains plenty of detail for those familiar with the statistics routinely employed in psychology but less for those with backgrounds in epidemiology and public health. Can the authors explain more about the effect sizes attributable to the intervention and their significance in practical terms?

We agree that additional information could ease interpretation of the statistical results. Therefore, we modified the Analyses section consequently: “Effect sizes were calculated using eta² (η²) and Cohen's d, with guidelines from Van den Berg (2022; η² = 0.01, small effect; η² = 0.06, medium effect; η² = 0.14, large effect) and Cohen (1988; d = 0.2, small effect; d = 0.5, medium effect; d = 0.8, large effect). Small, medium and large effect size indicates that the intervention had a minimal, moderate and substantial impact on the variability in doctoral mental health, respectively.” (p.9, l.227-231).

We also revised the Results section accordingly: “Results shown in Table 3 reveal significant Time * Group interactions for all variables (all p values <.006), with medium to large effects (all ηp2 values <.08). Further decompositions of these interactions revealed no significant differences between the pre-test means of the CG and EG for all variables (all p values >.59, all d values <.11). In contrast, substantial variations emerged when comparing the post-test means of the CG and EG (all p values <.007, all d values >.68). Specifically, a remarkable decline in psychological distress and an important increase in doctoral psychological, emotional, and social wellbeing from the pre-test to the post-test were observed for the EG (all p values <.001, all d values >.71), in stark contrast to the CG (all p values >.06, all d values <.38), as visually represented in Figure 4 below. Interestingly, while the positive effects of the retreat were moderate for doctoral social wellbeing (d = .71), the effects on psychological distress (d = 1.57) and emotional (d = 1.84) and psychological (d = 1.55) wellbeing were notably larger. These findings provide strong empirical support regarding the benefits of writing retreats in improving various facets of doctoral researchers’ mental health.” (p.10, l.368-381).

7. I can see why the authors wait until page 10 to elaborate on the second repeated measures analysis (using the intervention measures of the “waitlist control group”). I’m not so sure all readers will grasp this on first read and wonder if it merits a few more words to explain what was done and why it was important. (The authors already explain that these results reinforce the outcomes of the initial analysis.)

We clarified the reasons why we opted for a waitlist control group by adding the following precision in the Method section:

“The inclusion of a waitlist control group serves several important purposes (American Psychological Association, 2023). Firstly, from an ethical perspective, it ensures that all participants, including those in the control group, have an opportunity to experience the benefits of the intervention. This approach aligns with principles of fairness and equity in research, reducing potential harm or disappointment for participants in the control group.

Secondly, from a statistical standpoint, this methodology enhances the robustness of the study's findings. By testing the replicability of the results through a second group undergoing the same intervention, it strengthens the confidence in the observed effects. If consistent results are obtained in both groups, it suggests that the findings are not merely due to chance or specific conditions, increasing the validity of the research outcomes.(p.4, l.146-155).

The Analyses section was also modified to state more clearly that the study protocol included a waitlist control group as secondary analysis :

“To examine the replicability of the results, a second set of repeated-measures analyses was conducted, using the post-test scores of the waitlist CG as a pre-intervention measure, and the post-intervention scores as the post-intervention measure.” (p.9, l.231-234).

8. The qualitative findings seemed a little underplayed in the Discussion. One thing such results illustrate is that not everyone falls neatly into the general pattern indicated by quantitative findings. Is this worth a comment? Public health research is awash with indices of central tendency but real people have a funny way of not keeping in step.

We fully agree with this comment. Thus, we added a passage in the Discussion section: “Nonetheless, qualitative data uncovered nuanced aspects of participants' experiences that quantitative data could not capture. Notably, not everyone may benefit from the same aspects of the retreat. It is worth considering the possibility that distinct participant profiles exist, each with their own motivations for attending. This variety in participants' experiences and motivations suggests that a one-size-fits-all approach to retreat interventions may not be optimal.” (p.15, l.503-508).

9. Btw, I couldn’t get through to the Diener site via the link in the manuscript, and two of the three questionnaire links on the OSF registry returned blanks. Maybe it’s just a bad internet day but it might be worth checking those links again.

Thank you for bringing this to our attention. The link does seem to not be working anymore. We replaced it for this one : http://labs.psychology.illinois.edu/~ediener/SPANE.html (p.8, l.289).

We also checked the OSF and all questionnaires worked for us.

10. It was a pleasure to read a careful and meaningful study.

We thank you for your encouraging comments and highly beneficial insights.